# Deep Unsupervised Fusion Learning for Hyperspectral Image Super Resolution

**DOI:** 10.3390/s21072348

**Published:** 2021-03-28

**Authors:** Zhe Liu, Yinqiang Zheng, Xian-Hua Han

**Affiliations:** 1Graduate School of Science and Technology for Innovation, Yamaguchi University, Yamaguchi 753-8511, Japan; a501wbu@yamaguchi-u.ac.jp; 2National Institute of Informatics, Tokyo 101-8430, Japan; yqzheng@nii.ac.jp

**Keywords:** hyperspectral image, super-resolution, unsupervised fusion learning, image priors

## Abstract

Hyperspectral image (HSI) super-resolution (SR) is a challenging task due to its ill-posed nature, and has attracted extensive attention by the research community. Previous methods concentrated on leveraging various hand-crafted image priors of a latent high-resolution hyperspectral (HR-HS) image to regularize the degradation model of the observed low-resolution hyperspectral (LR-HS) and HR-RGB images. Different optimization strategies for searching a plausible solution, which usually leads to a limited reconstruction performance, were also exploited. Recently, deep-learning-based methods evolved for automatically learning the abundant image priors in a latent HR-HS image. These methods have made great progress for HS image super resolution. Current deep-learning methods have faced difficulties in designing more complicated and deeper neural network architectures for boosting the performance. They also require large-scale training triplets, such as the LR-HS, HR-RGB, and their corresponding HR-HS images for neural network training. These training triplets significantly limit their applicability to real scenarios. In this work, a deep unsupervised fusion-learning framework for generating a latent HR-HS image using only the observed LR-HS and HR-RGB images without previous preparation of any other training triplets is proposed. Based on the fact that a convolutional neural network architecture is capable of capturing a large number of low-level statistics (priors) of images, the automatic learning of underlying priors of spatial structures and spectral attributes in a latent HR-HS image using only its corresponding degraded observations is promoted. Specifically, the parameter space of a generative neural network used for learning the required HR-HS image to minimize the reconstruction errors of the observations using mathematical relations between data is investigated. Moreover, special convolutional layers for approximating the degradation operations between observations and the latent HR-HS image are specifically to construct an end-to-end unsupervised learning framework for HS image super-resolution. Experiments on two benchmark HS datasets, including the CAVE and Harvard, demonstrate that the proposed method can is capable of producing very promising results, even under a large upscaling factor. Furthermore, it can outperform other unsupervised state-of-the-art methods by a large margin, and manifests its superiority and efficiency.

## 1. Introduction

In hyperspectral (HS) imaging, three-dimensional cubic data with decades or hundreds of wavelength bands are captured. Each spatial point (pixel) contains a high dimensional vector for recording the light intensity at different wavelengths. Thus, the images acquired using the HS imaging technology contain not only abundant spatial structures but also detailed spectral signature and well suited to the substantial and high-performance analysis of imaged scenes. Having the advantages of detailed spectral distribution, the HS images were successfully used in various applications, such as remote sensing [1], food inspection [2,3,4], image classification [5,6,7] and object detection [8,9,10], and medicine [11,12,13], and are capable of achieving high-performance gain compared with other common RGB images. However, due to the radiant collection for each narrow-spectrum band in HS imaging sensors, less radiant energy per pixel and per spectral band measurement of an image scene is anticipated compared with the RGB imaging sensors. To ensure sufficient signal-to-noise ratio, the photo collection must be conducted in a much larger spatial region. This implies that the spatial resolution must be sacrificed to obtain detailed spectral information. Therefore, there is a trade-off between spatial and spectral resolution in real imaging sensors. This means that an HS sensor usually captures low spatial resolution and detailed spectral distribution (high spectral resolution) images. In contrast, common RGB sensors can provide much higher spatial resolution images but only with RGB color information. There are still some difficulties in acquiring high-resolution data in both spatial and spectral domains from commercial imaging sensors. Therefore, extensive research is necessary to fuse a low-resolution HS image (LR-HS) with the corresponding HR-RGB (multispectral) image for generating an HR-HS image using image processing and machine learning techniques. These fusion methods for generating HR-HS images are in general referred to as hyperspectral image super-resolution (HSI SR) methods [14].

Due to its ill-posed nature (the unknown number of variables in a latent HR-HS image is much larger than the known number of variables in the observations), multispectral and hyperspectral image fusion is a very challenging task. Most previously reported methods generally leverage various hand-crafted image priors to regularize the mathematical degradation model between observations and the latent HR-HS image. They also explore different optimization strategies to achieve the optimal solution. Based on the physical properties of the observed spectrum in HS images, one research direction is the investigation of effective representation of high-dimensional spectral vectors such as matrix factorization and spectral unmixing [15]. The spectral representation models were evolved from the fact that HS observations can generally be expressed as a weighted linear combination of the reflectance function basis and their corresponding fraction coefficients [16]. These models achieve acceptable reconstruction performance. Subsequently, based on the significant success of sparse representation in natural image processing, several research studies imposed a sparsity constraint on the spectral representation [17] as prior knowledge, and attempted to model the spatial structure and local spectral characteristics by automatically learning the spectral dictionary from the observed HR-RGB and LR-HS images. Additionally, based on possible low-dimensionality of the spectral space, a low-rank image prior technique was also adopted for exploring the intrinsic spectral correlation of a latent HR-HS image. This technique proved capable of reducing the spectral distortion to some extent  [18]. Moreover, some recent research studies extensively exploited the similarity between image priors of global spatial structures and local spectral structures to further boost reconstruction performance [19,20]. Although the integration of various hand-crafted image priors, such as physical spectral mixing, mathematical sparsity of spectral representation, low-rank property, and similarity resulted in significant progress regarding the HSI SR performance gain, discovering an optimal image prior for a specific scene is still an extremely difficult task due to the configuration and texture diversity in both the spatial and spectral domains.

The successful application of deep convolutional neural networks (DCNN) in various computer vision tasks allowed HSI SR to rely on the DCNN’s powerful learning capabilities [21] for robustly reconstructing a latent HR-HS image, and demonstrated impressive performance using various neural network architectures [22]. In contrast to the traditional optimization methods, the deep learning (DL) method is capable of automatically learning the underlying image priors in a latent HR-HS image instead of exploiting hand-crafted image priors. Using previously collected training samples consisting of LR-HS, HR-RGB images, and their corresponding HR-HS images, an HS image prediction model in the training phase can be constructed, and the corresponding HR-HS image can be efficiently estimated from its low-quality observations. However, most of the current DL methods are implemented in a fully supervised manner. For optimal network parameter learning, large-scale training triplets must be collected in advance. However, this is a difficult task, especially in the HSI SR scenario due to the high-cost of capturing the HR-HS images. Moreover, the fully supervised DL paradigm usually suffers from insufficient generalization in real applications and separate HS image prediction models for different HS datasets must be learned. More recently, Ulyanov et al.  [23] proposed a deep image prior (DIP) learning neural network and stated that a convolutional neural network itself is capable of capturing a large number of low-level image statistics for well reconstructing a natural image, which can be successfully applied to several image restoration tasks. Sidorov et al. [24] extended the idea of DIP into deep hyperspectral prior (DHP), and adopted denoising, inpainting and super-resolution for hyperspectral images. However, the DHPs only leverage the observed LR-HS image for network training and cannot efficiently learn both the spatial structure and spectral attribute priors for reconstructing a latent HR-HS image.

To deal with the limitations of current DL-based methods, a novel deep unsupervised fusion-learning (DUFL) framework for HSI SR is proposed in this work. Specifically, a generative neural network capable of automatically learning the spatial and spectral image priors in a latent HR-HS image is exploited. Instead of directly searching the target (HR-HS image) space, we attempt to determine the optimal parameters by searching the neural network parameter space to minimize the reconstruction errors of the observed HR-RGB and LR-HS images using a degradation procedure. For a robust neural network training to avoid falling into a local minimum state, the fixed initialized noisy input is perturbed with a small randomly generated noise at each training step, and the HR-HS image is predicted by employing the deep-learned network and the fixed noisy input. Moreover, special convolutional layers are designed for implementing the blurring/down-sampling operation and spectral transformation in the degradation model. By employing the deep-learned spatial and spectral image priors, the proposed DUFL method is capable of efficiently recovering the underlying spatial structure and spectral attribute of a latent HR-HS image using only the observed HR- RGB and LR-HS images. Also, massive triplets of the HR-RGB, LR-HS, and HR-HS images are not required. Experiments on two benchmark HS datasets demonstrate that the proposed framework is capable of achieving superior HS image reconstruction performance compared with most state-of-the-art (SoTA) methods in terms of both quantitative evaluation and visual effect. In summary, our main contributions are three-fold and summarized as follows:Propose a novel deep unsupervised fusion-learning framework, which can effectively learn the potential HR-HS image from the observed LR-HS and HR-RGB images only, for HSI SR problem;Leverage to search the parameter space of an encoder/decoder generative neural network instead of the raw target (HR-HS image) space to well reconstruct a latent HR-HS image, which has the prospects to learn both spatial structure and spectral attribute image priors;Construct an end-to-end unsupervised fusion-learning framework for the HSI SR problem by designing special convolutional layers for implementing the mathematical degradation model between the latent HR-HS image and the observations.

The rest of this paper is organized as follows. The related research work, including traditional optimization methods and deep-learning-based methods, is reviewed in Section 2. The proposed deep unsupervised fusion-learning framework is introduced in Section 3. Extensive experiments and comparison of results with two benchmark datasets are presented in Section 4. The concluding remarks are summarized in Section 5.

## 2. Related Research Work

In recent years, hyperspectral image reconstruction has been actively investigated in the computer vision and computational photography research community, and substantial improvement has been achieved. This work mainly focuses on hyperspectral image super resolution (HSI SR) by fusing the available LR-HS and HR-RGB images obtained from commercial imaging sensors. In this section, the related research work is briefly reviewed.

### 2.1. Traditional Optimization Method

The HSI SR problem, following the fusion paradigm of the observed LR-HS and HR-RGB images (fusion-based HSI SR), is closely related to a multi-spectral (MS) image pan-sharpening task, where the goal is to merge an LR-MS image with its corresponding HR wide-band panchromatic image [25]. Numerous methods for MS pan-sharpening, mainly including multi-resolution analysis approaches [26] and component substitution methods [27], were proposed. The fusion-based HSI SR problem can be treated as several pan-sharpening sub-problems, where each band of the HR-MS (RGB) image can be considered to be a panchromatic image. However, this simplification cannot fully use the spectral correlation and usually leads to significant spectral distortion in the recovered HR-HS image.

Recently, existing fusion-based HSI SR methods have widely leveraged the hand-crafted image priors in a latent HR-HS image for robustly solving the inverse optimization problem. The investigated image priors play a key role in obtaining a plausible solution in the optimization problem. The popularly used image priors are mainly used to explore the hidden knowledge in spatial and spectral representation such as physical spectral mixing, sparsity, low-rank, and similarity [28]. Yokoya et al. [15] proposed to decompose the latent HR-HS image into a non-negative end-member matrix and an abundance matrix called negative matrix factorization (NMF). Then, they exploited a coupled NMF version (CNMF) to fuse a pair of HR-MS and LR-HS images, whereas Lanaras et al. [29] used a proximal alternating linearized-minimization method to optimize the coupled spectral unmixing model for HSI SR. Subsequently, the sparsity reguralized decomposition was extensively investigated by imposing sparse constraints on the abundance matrix [30]. Akhtar et al. [31] proposed a Bayesian sparse representation scheme, aiming to infer the underlying probability distributions of spectra and their proportions, which were decomposed from the latent HR-HS image. Grohnfeldt et al. [32] employed a joint sparse representation for separately modeling the spatial structure (local patch) in each individual band image. Based on the inherent low-dimensionality of the spectral space and the 3D structure in a latent HR-HS image, tensor factorization and low-rank image priors were actively integrated for the HSI SR problem [33], and the feasibility of the reconstructed HR-HS image was demonstrated. More recently, Dong et al. [34] considered the spatially non-local similarity of a latent HSI and proposed a non-negative structured sparse representation (NSSR) method. Han et al. [35] further integrated both the global spatial and local spectral similarity to boost reconstruction performance. Although these hand-crafted image prior algorithms demonstrated impressive performance, finding a suitable image prior for a specific scene is still a challenging task.

### 2.2. Deep Learning Based Methods

Based on the successful application of deep convolutional neural network in nature RGB image super-resolution, deep learning was also investigated for fusion-based HSI SR tasks. Instead of exploiting the hand-crafted image priors, in this method, the inherent image priors hidden in a latent HR-HS image are automatically learned, and a superior reconstruction performance can be achieved. The current fusion-based HSI SR employing deep learning is mainly divided into the fully supervised learning method and the unsupervised learning method.

#### 2.2.1. Fully Supervised Learning Method

To train the HS image prediction model, it is necessary to previously collect the training triplets, including the LR-HS, HR-RGB images, and corresponding ground-truth (label), i.e., the HR-HS images in the fully supervised method, and provided an elaborate design for fusing multiple modalities of observations with different spatial and spectral structures. Han et al. [36] conducted an initial investigation by directly inputting the fused data of an HR-RGB image and an up-sampled LR-HS image to a simple 3-layer CNN, and used a more complex CNN with a residual structure to achieve better performance [37]. Palsson et al. [38] explored a 3D CNN-based MS/HS fusion scheme, and integrated the principal component analysis (PCA) to reduce the computational cost. Dian et al. [39] proposed a multi-stage method, and used a simple optimization strategy for initial HS reconstruction and final refinement, while adopting a 20-layer CNN for learning a latent HR-HS image from the initial one. More recently, Wang et al. [40] exploited an efficient hyperspectral image fusion network by iteratively integrating the representation relations between the target and observations into the deep-learning network to achieve superior performance. Despite the high performance gain, these methods fail to generalize well among different datasets, and they need to separately train the reconstruction models regarding the datasets under investigation, even for small imaging condition changes.

#### 2.2.2. Unsupervised Learning Method

To deal with the generalization limitation in the fully supervised learning method, the unsupervised neural network was proposed as a good solution to the HSI SR problem. Qu et al. [41] investigated an unsupervised encoder-decoder architecture to solve the MS/HS fusion problem. With the potential applicability of using a CNN-based end-to-end unsupervised neural network, this method needs to be carefully optimized for the two sub-networks in an alternating way, and its performance can be further improved. Based on the deep image prior (DIP) and the fact that the convolutional neural network itself can capture a large number of low-level image statistics to achieve a well-reconstructed natural image, Sidorov et al. [24] extended the DIP concept for automatically learning the underlying priors for HS images (DHPs) and applied it to the spatial resolution enhancement of an hyperspectral image. However, the DHPs can only leverage the observed LR-HS image for network training and cannot efficiently learn both the spatial structure and spectral attribute image priors for reconstructing a latent HR-HS image. Furthermore, Zhang et al. [42] leveraged the generated training triplets (the LR-HS, HR-RGB, and HR-HS images) with different degradation models to learn a common deep model for predicting an initial HR-HS image, and then exploited unsupervised adaptation learning for fine-turning the initial estimation and automatically learning the degradation operations of the under-studying observations. Although remarkable performance gain has been achieved with different degradation models compared with most state-of-the-art methods, the performance of the fine-turning HR-HS image in the adaptation learning is greatly affected by the initially estimation in the common model, and is easy to fall into a local minimum solution. In addition, Nie et al. [43] proposed two steps of learning method, where the spatial and spectral degradation models were first predicted via modeling the relation between the HR-HS image and the observations: LR-HS and HR-RGB images, and then the latent HR-HS image is reconstructed with the previously estimated degradation models. Though, feasibility and potential of the HR-HS image super resolution with unsupervised strategy is verified, current methods usually explore different steps of learning for obtaining acceptable performance for this challenge unsupervised learning. This study exploits a generative network for predicting the underlaying priors in the latent HR-HS image and the simply modified versions of the vanilla constitutional layer for approximating the degradation models, and thus implement the unsupervised HR-HS image super resolution in one stage. Our proposed method is mostly related to the DHP method, although there is a substantial difference between them. In this work, we effectively adopt both the observed LR-HS and HR-RGB images and propose an unsupervised fusion-learning framework for more robust HS image reconstruction.

## 3. Deep Unsupervised Fusion-Learning Method

In this section, the mathematical formulae of the fusion-based HSI SR problem are initially explained. Then the proposed deep unsupervised fusion-learning method, including the generative neural network architecture for automatically learning the underlying priors of a latent HR-HS image and the implementation of the constructed loss function for network training are introduced.

### 3.1. Problem Formulation

Given the observed image pair (LR-HS image $X\in R^{w\times h\times L}$ and HR-RGB image $Y\in R^{W\times H\times 3}$), where *w* and *h* represent the width and height, the goal of HSI SR is to recover a HR-HS image $Z\in R^{W\times H\times L}$, where *W* and *H* represent the width and height of Y and Z, and *L* is the number of spectral channels in the HR-HS image. Generally, the mathematical relation for formulating degradation operations between the observed images: **X**, **Y** and the target HR-HS image **Z** can be expressed as follows:(1)$$X=k^{\left(\mathrm{Spa}\right)}⊗Z^{\left(\mathrm{Spa}\right)}↓+n,Y=Z*C^{\left(\mathrm{Spec}\right)}+n,$$
where ⊗ denotes the 2D convolution operator, $k^{\left(\mathrm{Spa}\right)}$ represents the spatial 2D blur kernel, (Spa)↓ is the spatial decimation (down-sampling) operator, $C^{\left(\mathrm{Spec}\right)}$ is the spectral sensitivity function of the RGB camera (three 1D spectral filters) for converting the L spectral band to the RGB band, and n is the additive white Gaussian noise (AWGN) of the noise level σ. The mathematical expression of the above degradation models can be expressed in the following simplified matrix format:(2)X=DBZ+n,Y=ZC+n,
where **B** and **D** stand for the blurring matrix in the spatial domain and down-sampling matrix, respectively, for transforming **Z** to **X**. **C** denotes the spectral sensitivity function (CSF) of an RGB sensor. Assuming that the degradation parameters **B**, **D**, and **C** (which can be obtained from the hardware design of the HS and RGB sensors) are known, a heuristic approach to intuitively minimize the following reconstruction errors using the observed **X** and **Y** for estimating **Z** is given as follows:(3)$$Z^{*}=\underset{Z}{\arg\min}\alpha \beta_{1}{\left||X-\mathrm{DBZ}|\right|}_{F}^{2}+\left(1-\alpha \right)\beta_{2}{\left||Y-\mathrm{ZC}|\right|}_{F}^{2},$$
where $\left|\right|\cdot {\left|\right|}_{F}$ stands for the Frobenium norm. Since the element numbers in the HR-RGB and LR-HS images are different, it generally needs to introduce the normalization weights such as $\beta_{1}=1/N_{1}$ and $\beta_{2}=1/N_{2}$, where $N_{1}$ and $N_{2}$ are the products of the pixel numbers and the spectral bands in LR-HS and HR-RGB images, respectively. Beside, we further exploit a hyperparameter α (0≤α≤1) to adjust the contributions between these two reconstruction errors. (Equation 3) aims to obtain an optimal $Z^{*}$ for minimizing the weighted reconstruction error of the observations. Using the assumed AWGN in (Equation 2), (Equation 3) is completely equivalent to maximize the likelihood of a latent HR-HS image given the observation of **X** and **Y**. It is known that in the HSI SR problem the total number of unknown variables in Z is much greater than the known variables in **X** and **Y**, and this constitutes a severely ill-posed problem. Recovering a robust HR-HS image based on the observations is an extremely difficult task. To overcome this problem, most existing methods explore various hand-crafted image priors to model the underlying structure of the HR-HS image, and then impose a regularization term on the reconstruction error minimization problem, which can be formulated as follows:(4)$$Z^{*}=\underset{Z}{\arg\min}\alpha \beta_{1}{\left||X-\mathrm{DBZ}|\right|}_{F}^{2}+\left(1-\alpha \right)\beta_{2}{\left||Y-\mathrm{ZC}|\right|}_{F}^{2}+\lambda \phi \left(Z\right),$$
where ϕ(Z) is a term for modeling the underlying structure of Z, and λ represents a hyper-parameter, which balances the regularization term and reconstruction error distributions. By introducing the prior probability function Pr(Z), where ϕ(Z)=−log(Pr(Z)), (Equation 4) can be explained as the widely used maximum a posterior (MAP) framework. Although high performance gain can be achieved using various hand-crafted image prior in the HSI SR scenario, finding an appropriate image priors for a specific scene remains a challenging task. This work aims to use the powerful learning capability of deep-learning networks for automatically learning the underlying image priors in latent HR-HS images. Based on a DIP work, where the deep network architecture itself possesses a large number of low-level image statistics (image priors), a generative network for learning the spatial and spectral priors in a latent HR-HS image is employed. Then a reliable HR-HS image constrained by the learned priors using only the low-quality observations is reconstructed.

### 3.2. Proposed Deep Unsupervised Fusion Learning Method

In this section, the deep unsupervised fusion-learning framework for the HSI SR problem is introduced. Recent deep-learning-based HS reconstruction methods proved that a DCNN is capable of effectively capturing the underlying spatial and spectral structures (common prior information) of latent HS images and demonstrated promising performance. However, these methods are generally implemented in a fully supervised manner, and require large-scale training triplets containing LR-HS, HR-RGB, and HR-HS images, which are difficult to be specifically collected for obtaining the training labels (HR-HS images). Extensive research on natural image generation (DCGAN [44] and its variants) proved that high-definition and high-quality images with some defined characteristics and attributes can be successfully generated from a random noisy input without the supervision of high-quality ground-truth. This indicates that the inherent structure (priors) of a latent image with the defined characteristics can be captured by searching the neural network parameter space, starting from a random initial state. Moreover, DIPs [23] were exploited to model the more specific structures of an under-studying scene with the guidance of its degradation version only and successfully applied to several natural image restoration tasks such as image de-noising, impainting, and super-resolution. In this paper, this un-supervision paradigm is followed, aiming to learn the specific spatial and spectral structures (priors) of a latent HR-HS image with the guidance of its degraded observations (LR-HS and HR-RGB images). The conceptual scheme of the proposed DUFL framework is illustrated in Figure 1. Specifically, a generative neural network $G_{\theta }$ (θ is the network parameter to be learned) is leveraged to model the underlying spatial and spectral structures of a latent HR-HS image Z. By replacing Z with $G_{\theta }$ in (Equation 4) and removing the regularization term ϕ(Z) due to the automatically captured priors in the generative network, the fusion-based HSI SR model can be reformulated as follows:(5)$$\theta^{*}=\underset{\theta }{\arg\min}\alpha \beta_{1}||X-\mathrm{DB}G_{\theta }\left(z_{\mathrm{in}}\right){\left|\right|}_{F}^{2}+\left(1-\alpha \right)\beta_{2}||Y-G_{\theta }\left(z_{\mathrm{in}}\right)C{\left|\right|}_{F}^{2},$$
where $z_{in}$ is the input of the generative neural network, and $G_{\theta }{\left(z_{in}\right)}_{i}$ represents the i-th element of the estimated HR-HS image. Instead of directly optimizing a raw HR-HS image, which is extremely large and not unique, (Equation 5) aims to search the parameter space of the generative neural network $G_{\theta }$ for leveraging the possessed priors in it. To solve the objective function using the searching parameter space of the generative neural network, a detailed description of the used input data and the substantial architecture of the generative neural network is provided. The implementation of the degradation operations, including the spatial blurring/down-sampling operation and spectral transformation, are also provided.

Input Data to the Generative Neural Network: A popular generative neural network is trained to create a target image with the predefined characteristics, where randomly sampled noisy vectors based on a distribution function, such as Gaussian or uniform distribution, are usually used as inputs to ensure sufficient variety and diversity in the generated images. In our HSI SR task, the corresponding HR-HS images of the observed degradation version (LR-HS and HR-RGB images) must be acquired. Therefore, an intuitive way is to adopt an initially sampled noisy vector $z_{in}^{0}$ and fix it to search the optimal network parameter space for a specific HR-HS image. However, the fixed noisy input possibly leads to the generative neural network falling into a local minimum state. This results in an un-plausible estimation of the latent HR-HS image. Thus, a perturbation in the fixed initial input with a small randomly generated noisy vector at each training step is proposed to avoid a local minimum state. The input vector for the i-th training step can be expressed as follows:(6)$$z_{in}^{i}=z_{in}^{0}+\beta n_{i},$$
where $n_{i}$ is the randomly sampled noise vector at the i-th training step, and β denotes the perturbation degree (a small scalar value). After learning the generative neural network $G_{\theta }$ with the perturbed input, a prediction using the fixed noisy vector $Z^{*}=G_{\theta }\left(z_{in}^{0}\right)$ as the final estimated HR-HS image is performed.

Detailed Architecture of the Generative Neural Network:An arbitrary DCNN architecture can be adopted to serve as the generative neural network $G_{\theta }$. Due to diverse information, such as salient structures, rich textures, and complex spectra in a latent HR-HS image, a generative neural network $G_{\theta }$ is required to provide sufficient modeling capabilities. Various generative neural networks, such as that in the adversarial learning scenario [Pix2pix and others], were already proposed and demonstrated their great potential to generate high-quality natural images [45]. In this work, an encoder-decoder architecture with its multi-level feature-learning property and simplification is leveraged, and the skip connections between the encoder and decoder paths are leveraged for feature reusing. A detailed generative neural network is illustrated in Figure 1. Both the encoder and decoder consist of 5 blocks, which can learn the representative features in different scales, and the outputs of all 5 blocks in the encoder side are transferred to the corresponding decoder with skip connection for reusing the extracted detailed features. Each block is composed of 3 convolutional layers, following the RELU activation function, where the max-pooling layer with a 2 × 2 kernel is used to reduce the feature map size between the encoder blocks, whereas the up-sampling layer is employed to doubly recover the feature map size between the decoder blocks. Finally, a convolutional output layer is adopted for estimating the latent HR-HS image. However, in the unsupervised learning setting, there is no ground-truth HR-HS image for guiding or evaluating the training states of the generative neural network. Then, the observed HR-RGB and LR-HS images are leveraged to construct the evaluation criterion described in (Equation 5).

Implementation of the Degradation Operations: From the predicted HR-HS images of the generative neural network, it is possible to use mathematical degradation operations to approximate the LR-HS and HR-RGB images and then formulate quantitative criteria (loss function) for network learning (Equation 5). However, mathematical operations outside the generative neural network may result in difficulties in the training procedure. In this work, without any loss of generalization, two parallel special convolutional blocks are leveraged to implement the degradation model following the generative neural network and construct an end-to-end learnable framework. Specifically, the vanilla convolution layer is modified to be adapted for approximating the blurring/down-sampling operations and spectral transformation. Since, in a real scenario, each spectral band undergoes the same blurring/down-sampling transformations, the same kernel is defined for different spectral bands (channels) in a depth-wise convolutional layer with the stride parameter as the spatial expanding factor, and the bias term is set to ‘False’. The formula for the blurring/dawn-sampling transformation can be expressed as follows:(7)$$\hat{X}=f_{\mathrm{DB}}\left(G_{\theta }\left(z_{in}^{0}\right)\right),$$
where $f_{\mathrm{DB}}\left(\right)$ represents the transformation of an especially designed depth-wise convolutional layer.

For implementing the spectral transformation from the generated HR-HS image $\hat{Z}$ to the approximation of X, a point-wise convolutional layer with 3 output channels is applied, and the bias term as set to ‘False’. The formula for this spectral transformation can be expressed as follows:(8)$$\hat{Y}=f_{C}\left(G_{\theta }\left(z_{in}^{0}\right)\right),$$
where $f_{C}\left(\right)$ represents the spectral transformation with the point-wise convolutional layer. Using these two parallel blocks, an end-to-end learnable framework can be realized. For the known spatial blurring/down-sampling degradation, the kernel weight of the especially designed depth-wise convolutional layer with the known ones is initialized, and the trainable parameter is set to ‘False’. Similarly, kernel weights are set for the point-wise convolutional layer as the known CSF (spectral transformation matrix) of the RGB camera. Therefore, the proposed fusion-learning framework is considerably flexible and can be easily adapted to various degradation models. Moreover, it also has the prospect of automatically learning the transformation parameters in the embedded convolutional blocks for unknown degradations, which is left for future research. By replacing the spatial and spectral degradation operations with the designed convolutional blocks, the loss function for training the proposed deep unsupervised fusion learning network can be rewritten as follows:(9)$$\begin{array}{c} \theta^{*}=\underset{\theta }{\arg\min}\alpha \beta_{1}||X-f_{\mathrm{DB}}\left(G_{\theta }\left(z_{\mathrm{in}}\right)\right){\left|\right|}_{F}^{2}+\left(1-\alpha \right)\beta_{2}||Y-f_{C}\left(G_{\theta }\left(z_{\mathrm{in}}\right)\right){\left|\right|}_{F}^{2}, \end{array}$$

The optimization of (Equation 9) for obtaining the optimal parameter set of the generative neural network can be considered to be a kind of “zero shot” learning [46]. During the training procedure, only the low-quality image pairs (that is, the observed LR-HS and HR-RGB images) are used without the corresponding label (the HR-HS images) and thus the proposed method is completely unsupervised of being generalized for any real observations. The detail implementation for the proposed DUFL method is summarized in Algorithm 1.
**Algorithm 1** Algorithm of the proposed deep unsupervised fusion learning method.**Input:** The observed LR-HS image X and HR-RGB image Y**Output:** Latent HR-HS image Z1: Sample $z_{in}^{0}$ from uniform distribution with seed 02: **for**
i=0 to max. iter. (I)
**do**3:     Sample $n_{\left(0,1\right)}^{i}$ from uniform distribution4:     Perturb $z_{in}^{0}$ with $n_{\left(0,1\right)}^{i}$: $z_{in}^{i}=z_{in}^{0}+\beta n_{\left(0,1\right)}^{i}$5:     $\hat{Z}=G_{\theta }\left(z_{in}^{i},\theta^{i-1}\right)$6:     $\hat{X}=f_{DB}\left(\hat{Z}\right)$7:     $\hat{Y}=f_{C}\left(\hat{Z}\right)$8:     Loss function: $\alpha \beta_{1}||X-\hat{X}{\left|\right|}_{F}^{2}+\left(1-\alpha \right)\beta_{2}||Y-\hat{Y}{\left|\right|}_{F}^{2}$9:     Compute the gradients regarding $G_{\theta }$10:     Update θ using the ADAM algorithm [47] as $\theta^{i}$11: **end for**12: $Z=G_{\theta }\left(z_{in}^{0}\right)$

## 4. Experiment Results

### 4.1. Experimental Settings

Datasets:Two benchmark HSI datasets, including the CAVE [48] and Harvard [49] datasets, were adopted to evaluate the efficiency of the proposed method. The CAVE dataset contains 32 HS images with a spatial resolution of 512 × 512 pixels and a spectral resolution of 31 bands in the spectral range 400 nm–700 nm. The images in the CAVE dataset contain various real-world materials and objects, such as human faces, fruit, etc. The Harvard dataset contains 50 images of various natural scenes with a spatial resolution of 1392 ×1040 pixels and a spectral resolution of 31 bands in the spectral range 420 nm–720 nm. We illustrate some example images of the CAVE and Harvard datasets in Figure 2. In the experiments conducted, the upper-left sub-image portions of size 1024 × 1024 pixels were cropped from the original HS images in the Harvard dataset and downsampled them to 512 × 512 pixel images, which constitute the ground-truth HS images. The observed LR-HS images were synthesized from the ground-truth HS images in both datasets using bicubic down-sampling with different spatial decimation factors (8, 16, and 32) to create images with a size of 64 × 64 × 31, 32 × 32 × 31 and 16 ×16 × 31, respectively. Meanwhile, the observed HR-RGB images were generated by multiplying the spectral response function of the Nikon D700 camera with the ground truth HR-HS images.

Comparison Methods: There are three main research paradigms for HS image super-resolution, containing (1) traditional optimization methods with manually engineered image priors based on accumulated experience or clarified physical knowledge; (2) unsupervised methods with automatically learned image priors; (3) fully supervised deep-learning methods with learned external image priors (training samples). First, the proposed method (unsupervised method with learned image priors) with a spatial up-scale factor of 32 of the CAVE dataset was compared with different paradigms of SoTA methods, including GOMP [50], MF [16], SNNMF [51], DHIP [24], uSDN [41], PNN [52], 3D-CNN [53], and CMHF-net [54] methods. Since the fully supervised deep-learning methods (PNN [52], 3D-CNN [53], and CMHF-net [54]) must be separately trained to construct individual models for different spatial up-scale factors of different datasets and a long time is required to train just one model, the results for other up-scale factors (8, 16) of the CAVE dataset and all factors of the Harvard dataset were compared with those obtained from the unsupervised SoTA methods.

Evaluation Metrics: To quantitatively evaluate the performance of the reconstructed HS images, the following five commonly used metrics were adopted: the root-mean-square-error (RMSE), peak signal-to-noise ratio (PSNR), structural similarity (SSIM), spectral angle mapper (SAM) and relative size global error (ERGAS). The RMSE, PSNR, and ERGAS were individually calculated per pixel and per spectral band and averaged over all spatial positions and spectral bands. The SSIM was computed for spatial structure evaluation. All these metrics exhibit the spatial fidelity of the reconstructed HSIs. The SAM exhibits spectral fidelity, which is calculated for each 1-D spectral vector and averaged over all spatial positions. Larger values of the PSNR and SSIM indicate better performance, whereas smaller values of the RMSE, ERGAS, and SAM imply better SR results.

Implementation Details: The proposed method was implemented in Pytorch. The input noise was initially set to the same dimension as the to-be-estimated HR-HS image. To train the generative network, the Adam optimizer [47] with the simple $L_{2}$ norm-based loss was adopted. The learning rate was initialized to 1e-3 and reduced by 0.7 every 1000 steps. Moreover, the perturbation degree β was initially set to 0.05 and reduced by 0.5 every 1000 steps. The optimization process was terminated after 12,000 steps and fixed for all images with different upscale factors from different datasets. All experiments were conducted in the learning environment with Tesla K80 GPU. In our experiment, the learning time for an image with size of 512 × 512 is about 20 min.

### 4.2. Numerical Results Comparison

Table 1 shows the compared quantitative evaluation of the super-resolved HS images for an up-scale factor of 32 of the CAVE dataset with different paradigms, including the traditional methods with manually engineered image priors (GOMP [50], MF [16], SNNMF [51]), the unsupervised methods with learned priors (DHIP [24], uSDN [41]), and fully supervised deep-learning methods (PNN [52], 3D-CNN [53], and CMHF-net [54]). Since an HR-HS image in the proposed method can be predicted by the generative neural network using the initial fixed noise w/o perturbation, two predictions with different inputs (denoted as DUFL and $DUFL_{+}$) can be achieved. In Table 1, it can be observed that the proposed method significantly outperforms the SoTA methods, belonging to the same unsupervised paradigm with the learned image priors (DHIP [24] and uSDN [41]). For the results obtained using the uSDN [41] method, the released code (https://github.com/mbaddeley/usdn, accessed on 18 October 2020), was re-run with default hyper-parameters, and only 8 samples out of 32 images provided correct super-resolved results. Thus, the average performance using only 8 correct samples is presented in Table 1. Although the proposed DUFL method can be robustly learned with the provided hyper-parameters mentioned in the previous Section, it is not necessary to adjust these parameters for different datasets and up-scale factors. It should also be noted that it is possible to improve its performance to conduct hyper-parameter turning for different datasets and up-scale factors. Moreover, the proposed method exhibits better performance than the fully supervised deep-learning methods and most traditional methods. One possible reason for this performance is the small number of samples used for training the HSI SR model, which is a challenge issue in the HSI SR scenario. The proposed method exhibits comparable results (without integrating any prior knowledge) with the SNNMF [51] method by leveraging the manually engineered image priors, which evolved and proved to be efficient by conducting a long-time research effort. Additionally, the proposed DUFL method aims to generate latent HR-HS images from the noise input and does not leverage any existing rich spectral information and spatial structure in the observed LR-HS and HR-RGB images for regularizing the generative neutral network learning. Thus, the HSI SR performance is expected to be improved by imposing some constraints on the network training using the observations. This is left for future investigation.

A comparison between the results obtained using the unsupervised SoTA methods on the CAVE dataset (with up-scale factors of 16 and 8) and the Harvard dataset is prensented in Table 2 and Table 3, respectively. In these tables, it can be observed that the proposed method achieves better or comparable results with those obtained using SoTA methods. Furthermore, it must be clarified that the uSDN method cannot produce accurate results for the same samples, especially for large up-scale factors, and the average evaluations computed only with the correct outputs are presented in Table 2 and Table 3. In Table 1, Table 2 and Table 3, the up-arrow indicates better result with larger value while the down-arrow denotes better result with smaller value. Meanwhile, in order to evaluate the effect of different data terms on the loss function, we set the hyper-parameter α value as 0.3, 0.5 and 0.7, respectively, and provide the compared result in Table 4. Table 4 manifests the quantitative metrics of PSNR, SAM and ERGAS with our DUFL+ method, which illustrates that there are no large impact on the super-resolution performance via fine-tuning the hyper-parameter α. We are going to investigate in detail this phenomenon in the future.

### 4.3. Perceptual Quality

To visualize the reconstruction results, two representative images from the CAVE and Harvard datasets, respectively, with different upscale factors (8, 16, and 32) are shown in Figure 3, Figure 4, Figure 5, Figure 6, Figure 7 and Figure 8 using different deep unsupervised methods (DHIP [24], uSDN [41], the proposed (DUFL method, and the $DUFL_{+}$ method w/o a perturbation term). In all figures, the first row shows the original HR image and the super-resolved results with spatial and spectral fidelity indexes (PSNR Sam for the recovered images using different methods), whereas the second row shows the difference images between the recovered HR-HS image and the ground-truth HR-HS image. In these figures, it can be observed that the proposed DUFL method is capable of recovering the HR-HS image with a smaller difference to the ground-truth HR-HS image and more reliable spatial/spectral indexes for most cases, excluding the results of uSDN in Figure 8. As mentioned in Section 4.2, although the uSDN method is capable of achieving impressive performance for some specific images using the default hyper-parameters, it cannot provide accurate results for most of the images, especially for those with large upscale factors. Despite the better results obtained for some images using the uSDN method, accurate recovered results cannot be achieved for 20 images out of the total 50 images. Moreover, the proposed $DUFL_{+}$ method is capable of further improving the performance of the DUFL-version method for all images with different upscale factors.

## 5. Conclusions

In this work, a deep unsupervised fusion-learning framework for the hyperspectral image super-resolution problem was proposed. Based on the fact that the neural network architecture itself can capture a large number of low-level image statistics, a deep convolutional neural network for automatically learning the underlying spatial and spectral image priors in latent HR-HS images from a perturbed noisy input was leveraged. Specifically, an encoder-decoder architecture with skip connection was adopted for modeling the diverse context in a latent HR-HS image, and the special depth-wise and point-wise convolutional blocks were designed to implement the degradation transformations between observations and the required target, thus establishing an end-to-end learnable framework using only low-quality observations. The proposed unsupervised learning neural network is capable of effectively leveraging the HR spatial structure in HR-RGB images and the detailed spectral properties in LR-HS images to provide more reliable HS image reconstruction without using any training samples. Extensive experiments on two hyperspectral datasets, including the CAVE and Harvard datasets, demonstrated that the proposed method is capable of achieving impressive performance in terms of quantitative evaluation and visual effect.

## Figures and Tables

**Figure 1 sensors-21-02348-f001:**
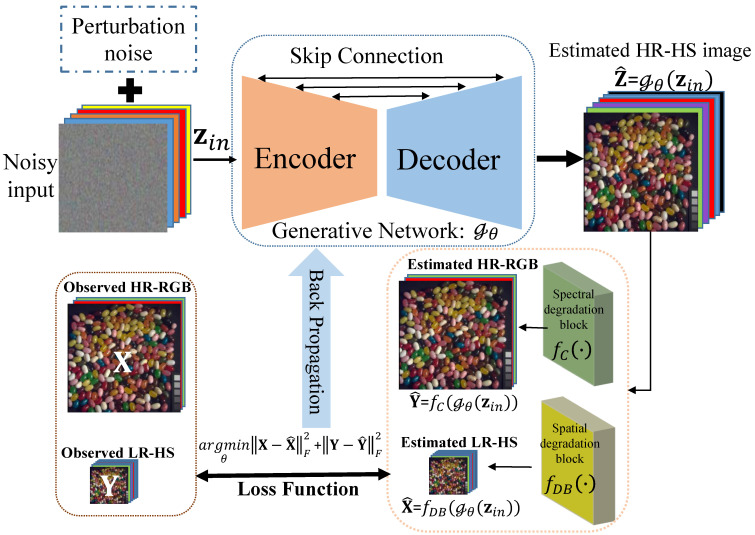
Conceptual diagram of the proposed DUFL method. An encoder-decoder-based generative neural network is used for modeling the rich spatial and spectral structures of a latent HR-HS image, and two parallel specifically designed convolutional blocks are used for approximating the spatial and spectral degradation models. Then the proposed framework is trained with the observations (LR-HS and HR-RGB images).

**Figure 2 sensors-21-02348-f002:**
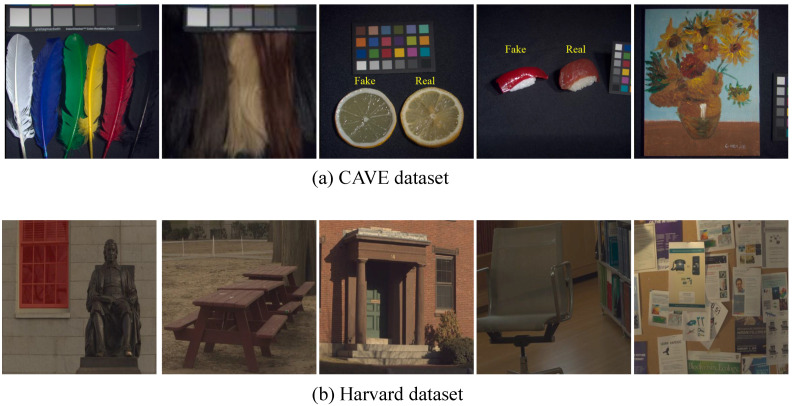
Example images in the CAVE and Harvard datasets.

**Figure 3 sensors-21-02348-f003:**
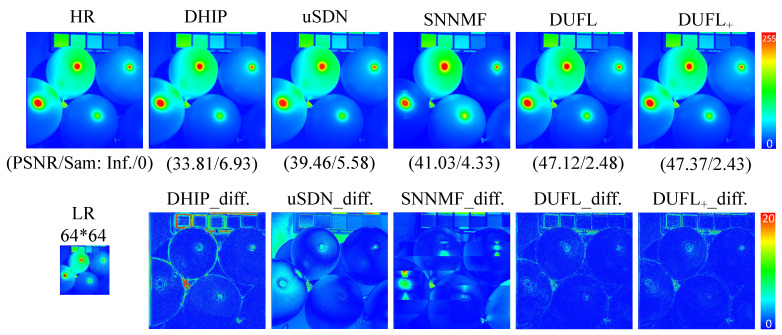
Recovered HR-HS image of the ${}^{‘}$Balloon${}^{’}$ sample in the CAVE dataset using the DHIP [24], uSDN [41], SNNMF [51], and the proposed methods and corresponding difference images between the ground-truth and recovered images with an upscale factor of 8.

**Figure 4 sensors-21-02348-f004:**
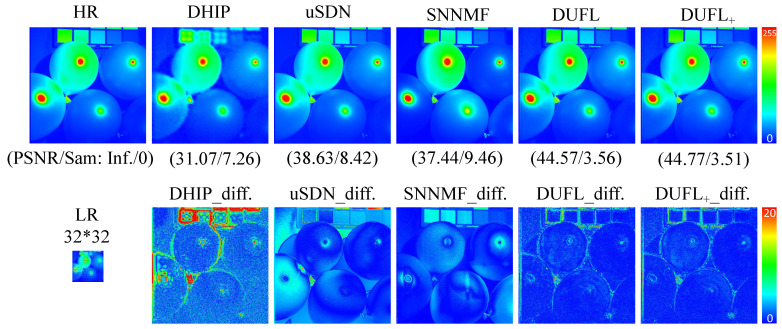
Recovered HR-HS image of the ${}^{‘}$Balloon${}^{’}$ sample in the CAVE dataset using the DHIP [24], uSDN [41], SNNMF [51], and the proposed methods and corresponding difference images between the ground-truth and recovered images with an upscale factor of16.

**Figure 5 sensors-21-02348-f005:**
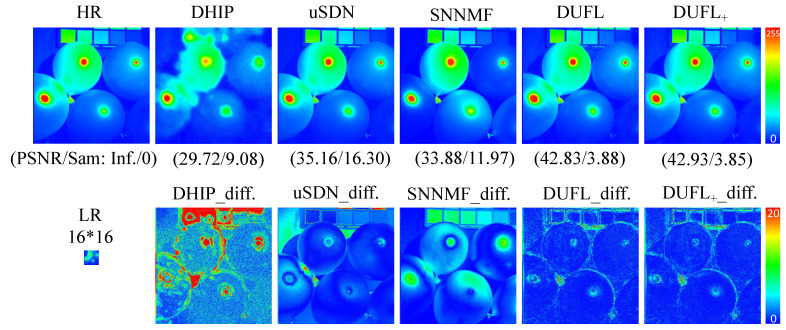
Recovered HR-HS image of the ${}^{‘}$Balloon${}^{’}$ sample in the CAVE dataset using the DHIP [24], uSDN [41], SNNMF [51], and the proposed methods and corresponding difference images between the ground-truth and recovered images with an upscale factor of 32.

**Figure 6 sensors-21-02348-f006:**
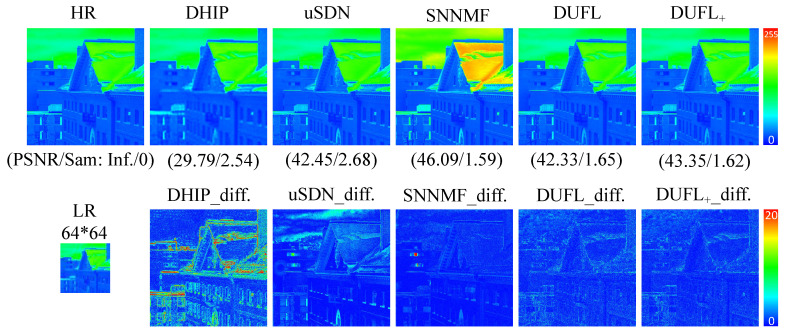
Recovered HR-HS image of the ${}^{‘}$img1${}^{’}$ sample in the Harvard dataset using the DHIP [24], uSDN [41], SNNMF [51], and the proposed methods and corresponding difference images between the ground-truth and recovered images with an upscale factor of 8.

**Figure 7 sensors-21-02348-f007:**
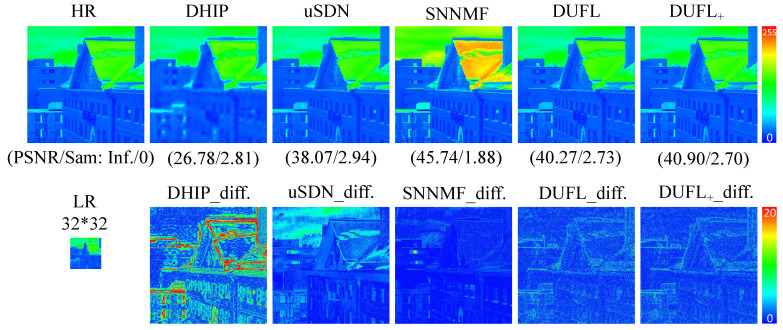
Recovered HR-HS image of the ${}^{‘}$img1${}^{’}$ sample in the Harvard dataset using the DHIP [24], uSDN [41], SNNMF [51], and the proposed methods and corresponding difference images between the ground-truth and recovered images with an upscale factor of 16.

**Figure 8 sensors-21-02348-f008:**
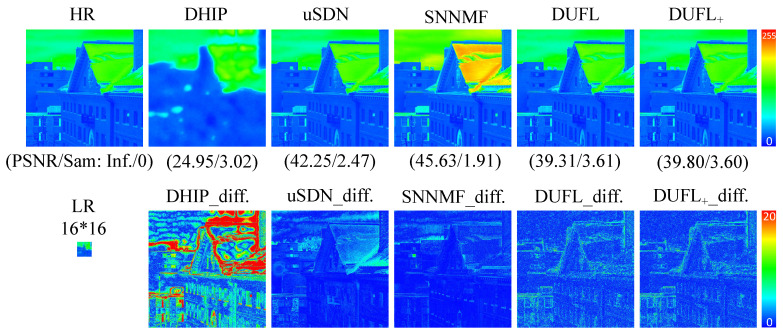
Recovered HR-HS image of the ${}^{‘}$img1${}^{’}$ sample in the Harvard dataset using the DHIP [24], uSDN [41], SNNMF [51], and the proposed methods and corresponding difference images between the ground-truth and recovered images with an upscale factor of 32.

**Table 1 sensors-21-02348-t001:** Comparison of the quantitative evaluation results obtained using the proposed method with those obtained by state-of-the-art methods, including traditional methods with manually engineered image priors (GOMP [50], MF [16], and SNNMF [51]), unsupervised methods with learned image priors (DHIP [24] and uSDN [41]), and fully supervised deep-learning methods (PNN [52], 3D-CNN [53], and CMHF-net [54]).

Type of Priors	Method	RMSE↓	PSNR↑	SSIM↑	Sam↓	ERGAS↓
Manually Engineered Priors	GOMP	6.47	32.48	-	14.19	0.77
	MF	3.03	39.37	-	6.12	0.40
	SNNMF	3.26	38.73	-	6.50	0.44
Learned Priors	uSDN	4.217	35.79	0.9216	15.75	0.5171
	DHIP	16.01	24.73	0.7449	13.0761	2.1490
	DUFL (Our)	3.47	38.17	0.9512	8.31	0.4597
	DUFL+ (Our)	3.34	38.47	0.9548	8.12	0.4436
Supervised External Priors	PNN	6.103	32.42	0.962	14.73	1.3451
	3D-CNN	4.629	34.82	0.937	8.96	1.0920
	CMHF-net	3.507	37.23	0.962	7.30	0.8187

**Table 2 sensors-21-02348-t002:** Comparison of the evaluation results using the proposed methods with those obtained using the unsupervised SoTA methods, including the manually engineered image priors (GOMP [50], MF [16], SNNMF [51]) and automatically learned image priors (DHIP [24] and uSDN [41]) in the CAVE dataset for up-scale factors of 16 and 8.

Up-Scale Factor	Type of Priors	Method	RMSE↓	PSNR↑	SSIM↑	Sam↓	ERGAS↓
16	Manually Engineered Priors	GOMP	6.08	32.96	-	12.60	43
		MF	2.71	40.43	-	4.82	0.73
		SNNMF	2.45	41.21	-	4.61	0.66
	Learned Priors	uSDN	4.389	35.74	0.9111	18.7928	1.3469
		DHIP	11.31	27.76	0.8024	10.6639	3.0838
		DUFL (Our)	2.6053	40.71	0.9665	6.6163	0.6990
		DUFL+ (Our)	2.497	41.03	0.9694	6.434	0.6741
8	Manually Engineered Priors	GOMP	5.69	33. 64	-	11.86	2.99
		MF	2.34	41.83	-	3.88	1.26
		SNNMF	1.89	43.53	-	3.42	1.03
	Learned Priors	uSDN	5.808	33.402	0.8779	24.2909	7.0549
		DHIP	7.5995	31.4040	0.8708	9.2542	4.2025
		DUFL (Our)	2.078	42.50	0.9750	5.355	1.155
		DUFL+ (Our)	1.957	42.98	0.9774	5.223	1.098

**Table 3 sensors-21-02348-t003:** Comparison of the evaluation results using the proposed methods with those obtained using the unsupervised SoTA methods, including the manually engineered image priors (GOMP [50], MF [16], SNNMF [51]) and automatically learned image priors (DHIP [24] and uSDN [41]) in the Harvard dataset for up-scale factors of 32, 16, and 8.

Up-Scale Factor	Type of Priors	Method	RMSE↓	PSNR↑	SSIM↑	Sam↓	ERGAS↓
32	Manually Engineered Priors	GOMP	4.08	38.02	-	4.79	0.41
		MF	1.96	43.19	-	2.93	0.23
		SNNMF	2.20	42.03	-	3.17	0.26
	Learned Priors	uSDN	3.328	37.75	0.9729	6.091	0.5162
		DHIP	13.25	26.23	0.7186	5.6758	1.4121
		DUFL (Our)	2.8203	40.119	0.9571	3.9550	0.4268
		DUFL+ (Our)	2.6199	40.7461	0.9627	3.9124	0.4173
16	Manually Engineered Priors	GOMP	3.83	38.56	-	4.16	0.77
		MF	1.94	43.30	-	2.85	0.47
		SNNMF	1.93	43.31	-	2.82	0.45
	Learned Priors	uSDN	2.409	40.529	0.9827	3.834	0.6308
		DHIP	10.38	38.44	0.754	4.57	2.08
		DUFL (Our)	2.814	40.767	0.9527	3.0081	0.7479
		DUFL+ (Our)	2.556	41.656	0.9593	2.9529	0.7118
8	Manually Engineered Priors	GOMP	3.79	38.89	-	4.00	1.65
		MF	1.83	43.74	-	2.66	0.87
		SNNMF	1.79	43.86	-	2.63	0.85
	Learned Priors	uSDN	2.423	42.11	0.9848	3.882	1.1707
		DHIP	7.9489	30.8609	0.8029	3.5295	3.1509
		DUFL (Our)	2.3842	42.16	0.9646	2.3545	1.0869
		DUFL+ (Our)	2.1167	43.23	0.9709	2.3045	1.007

**Table 4 sensors-21-02348-t004:** Comparison of the evaluation results using our proposed method: DUFL+ with different weights α values of 0.3, 0.5 and 0.7 in the CAVE and Harvard datasets for up-scale factors of 16 and 8. Bold values indicate the best results.

		CAVE			Harvard		
**Up-Scale Factor**	α	**PSNR**↑	**SAM**↓	**ERGAS**↓	**PSNR**↑	**SAM**↓	**ERGAS**↓
32	0.3	38.8653	7.0733	0.3312	39.4625	3.8458	0.4368
	0.5	38.0345	7.2645	0.3708	40.0153	3.5062	0.3910
	0.7	39.1060	6.6191	0.3334	39.0670	3.3812	0.3879
16	0.3	40.7483	5.7067	0.5395	40.9471	2.8952	0.6622
	0.5	40.7500	5.8747	0.5518	40.7912	2.7029	0.6181
	0.7	40.4173	5.6399	0.5829	41.9040	2.4800	0.5177
8	0.3	42.1921	5.0925	0.9452	43.0742	2.1642	0.9310
	0.5	42.9066	4.4038	0.8570	41.6796	2.1902	1.0589
	0.7	42.1635	4.7505	0.9171	41.8517	2.1843	1.0885

## Data Availability

Publicly available datasets were analyzed in this study. The data can be found here: [https://www.cs.columbia.edu/CAVE/databases/multispectral/] (accessed on 28 March 2021) and [http://vision.seas.harvard.edu/hyperspec/d2x5g3/] (accessed on 28 March 2021).

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
