# Peer review of "Deep Unsupervised Fusion Learning for Hyperspectral Image Super Resolution"

_sensors, 2021, doi:10.3390/s21072348_

Round 1

Reviewer 1 Report

The manuscript introduces a method for fusing a HR-LR and a LR-HS image into a HR-HS one. This method is based on CNN and, as a novelty, is able to natively deal with multi- (hyper)-spectral images.

The topic is interesting, and the paper is well written, with a very good Introduction, and methods are clearly presented. Therefore I believe that it can be considered for publication in the joiurnal.

I'm asking to address the following minor items:

  • In Eqs. (3-5) it would be more standard to use || ... || for a Frobenius norm instead of | ... |
  • In Eqs (3-5) the two terms for X and Y are not weighted (equivalently they have the same weight 1). However according to the definition of the norm, the actual weight depends on the size of X and Y, that is on the number of pixels and the number of channels for both of them. It would be possible to make the weight of the two terms insensitive of the number of pixels and channels by normalizing them by the square root of the product of the corresponding number of pixels x number of channels. There would be any change/improvment of the results?
  • Tables 1, 2 and 3: it is not clear how figures are selected to be written as boldface

Some misprints:

  • l. 74-75: double recent
  • l. 100: maybe stated instead of started
  • l. 100: capable instead of "a capable"
  • l. 127: follows instead of followa
  • l. 167-168: decompose instead of "the decomposition"
  • l. 239: "Generally the mathematical relation for formulating degradation operations between the observed images: X": please restructurate(there is n verb int he sentence)
  • l. 239 (line before formula (2): "can be expressed" instead of "can be"
  • l. 243: widely instead of widerly
  • l. 245: prior instead of priors
  • l. 253: rephrase the sentence "where G_θ (θ is the network parameter to be learned), to model..."
  • l. 270: "to provide" instead of "toprovide"
  • l. 397: is instead of ia

Reviewer 2 Report

In this paper, the authors present a hyperspectral image fusion approach based on Deep Learning. The context of the problem is correctly presented in the introduction. Then, a description of the state-of-art methods for hyperspectral image fusion is reviewed; and the proposed method is clearly explained.  Some comments about the manuscript:   - As far as I am concerned, in the literature the term superresolution is used in other contexts. I think what the authors are proposing is an image fusion technique. - The English usage should be carefully revised prior to publication. - The manuscript template is not adequate. - References 2-5, regarding hyperspectral imaging applications, can be updated.  - I would suggest the authors to include more references related to the use of unsupervised DL for hyperspectral image fusion. Then, I think the authors should clearly state the main differences of the proposed method compared to such references. - Regarding Section 4.1, why did the authors downsample the Harvard dataset to 512x512? - I would suggest the authors to improve the contents of Section 4.2. Besides, I think the Tables can be integrated with the text to improve the overall readability of the document. Additionally, I would suggest to clarify which criteria was used to bold the results in Tables1-3. In the current version such bold values are confusing for me. - I would also suggest to re-arrange the document in order to find an appropriate place for Figures 2-7.   According to my comments, I think the research presented in this manuscript has value. However, from my point of view, the authors should address the aforementioned comments prior to its publication.

Reviewer 3 Report

This paper proposes a deep unsupervised fusion learning (DUFL) method, for high resolution hyperspectral image reconstruction. In this method, the spatial and spectral image priors can be automatically learned through a generative neural network, instead of using pre-defined priors, which greatly increases the practicality in real applications. The experiments conducted on two real hyperspectral datasets show the promise of the effectiveness of DUFL.

The motivation and background are well-introduced. The methodology part and theoretical part sound reasonable. The related work is well introduced, too. The reviewer thinks this paper provides significant contribution which is potential to be published. However, there are few questions/issues encouraged to be fixed. The comments are as follows:

Major comments:

  1. Since the approach is somehow complicated, the authors are encouraged to add a step diagram or an algorithm, to describe the detail procedure of the proposed DUFL in the end of Section 3. Doing this can make the readers understand the implementation of DUFL.

  1. The network structure of the generative neural network (encoder-decoder) is not explained. Is it the exiting ED (such as U-Net) or a newly designed one?

  1. What GPU do the authors use for training the generative network? And how about the training time? Please provide information about computational complexity in the revised version.

  1. In Tables 1~3, what do the bold values mean? Please define. In general, the bold values indicate the best performance. However, in Table 3 the uSDN’s values are all bold. The Table 4 almost misses the bold values.

  2. It is encouraged to show the sample images of CAVE and Harvard dadastes in Section 4.1.

  3. Figs 2-7 demonstrate the “difference” images of ground truth HR-HS images and DUFL-recovered ones. The reviewer feels curious, why don’t the authors directly demonstrate the recovered images along with the ground truth images? Showing original image will be more intuitive than showing difference image. And we can compare the pros/cons of different categories of methods (manually engineered priors methods and Learned priors methods)

Minor comments:

  1. In Abstract, the abbreviation “HS” is not defined.

  1. In line 75, there are two “recent”.

  1. Line 127. followaàfollow.

  1. In page 8, before Eq(5), HSR SR à HSI SR ?

  1. Line 270, toprovideà to provide

  1. In Figure 3, balloon à balloon?

  1. In Fig.1, the term LR-LS à LR-HS?

Round 2

Reviewer 3 Report

The authors have addressed all the issues that the reviewer concerned.